# Urotensin II Enhances Advanced Aortic Atherosclerosis Formation and Delays Plaque Regression in Hyperlipidemic Rabbits

**DOI:** 10.3390/ijms24043819

**Published:** 2023-02-14

**Authors:** Qingqing Yu, Panpan Wei, Liran Xu, Congcong Xia, Yafeng Li, Haole Liu, Xiaojie Song, Kangli Tian, Weilai Fu, Rong Wang, Weirong Wang, Liang Bai, Jianglin Fan, Enqi Liu, Sihai Zhao

**Affiliations:** 1Institute of Cardiovascular Science, Translational Medicine Institute, Xi’an Jiaotong University Health Science Center, Xi’an 710061, China; 2Laboratory Animal Center, Xi’an Jiaotong University, Xi’an 710061, China; 3Department of Molecular Pathology, Faculty of Medicine, Graduate School of Medical Sciences, University of Yamanashi, Tokyo 409-3898, Japan; 4Department of Cardiology, The Second Affiliated Hospital of Xi’an Jiaotong University, Xi’an 710004, China

**Keywords:** urotensin II, atherosclerosis, vulnerable plaque, angiogenesis, coronary plaque

## Abstract

Accumulated evidence shows that elevated urotensin II (UII) levels are associated with cardiovascular diseases. However, the role of UII in the initiation, progression, and regression of atherosclerosis remains to be verified. Different stages of atherosclerosis were induced in rabbits by a 0.3% high cholesterol diet (HCD) feeding, and either UII (5.4 μg/kg/h) or saline was chronically infused via osmotic mini-pumps. UII promoted atherosclerotic fatty streak formation in ovariectomized female rabbits (34% increase in gross lesion and 93% increase in microscopic lesion), and in male rabbits (39% increase in gross lesion). UII infusion significantly increased the plaque size of the carotid and subclavian arteries (69% increase over the control). In addition, UII infusion significantly enhanced the development of coronary lesions by increasing plaque size and lumen stenosis. Histopathological analysis revealed that aortic lesions in the UII group were characterized by increasing lesional macrophages, lipid deposition, and intra-plaque neovessel formation. UII infusion also significantly delayed the regression of atherosclerosis in rabbits by increasing the intra-plaque macrophage ratio. Furthermore, UII treatment led to a significant increase in NOX2 and HIF-1α/VEGF-A expression accompanied by increased reactive oxygen species levels in cultured macrophages. Tubule formation assays showed that UII exerted a pro-angiogenic effect in cultured endothelial cell lines and this effect was partly inhibited by urantide, a UII receptor antagonist. These findings suggest that UII can accelerate aortic and coronary plaque formation and enhance aortic plaque vulnerability, but delay the regression of atherosclerosis. The role of UII on angiogenesis in the lesion may be involved in complex plaque development.

## 1. Introduction

Cardiovascular and cerebrovascular diseases remain the first cause of global human death [1,2]. Atherosclerosis is the most important pathological bases of cardiovascular and cerebrovascular diseases, including myocardial infarctions, strokes, and disabling peripheral artery disease [3]. Urotensin II (UII), a cyclic peptide of 11 amino acids that was first discovered in teleost fish, is also linked to the pathophysiology and physiology of the heart [4,5]. Previous studies identified a G-protein-coupled orphan receptor 14 (GPR14) as the UII receptor (UTR) [6,7]. Increased plasma UII levels have been linked to cardiovascular diseases, such as in patients who first experience acute coronary syndrome (ACS) symptoms [8]. UII could be considered an important biomarker for atherosclerotic cardiovascular diseases (ASCVD). In patients with ASCVD, the UII plasma levels in ACS patients were higher than the healthy control subjects [9,10,11,12].

UII may be involved in the regulation of all stages of atherosclerotic cardiovascular diseases. Previously, we reported that UII can enhance atherosclerosis initiation by upregulating cell adhesion molecules (CAM), including ICAM-1 and VCAM-1 of vascular endothelia, to promote leukocyte cell adhesion [10,13]. However, the role of UII in advanced and complicated atherosclerotic plaque formation has not been investigated. This is an important issue because the formation and progression of vulnerable coronary plaques often lead to ACS and threaten life [14,15,16]. It has been reported that a thin fibrotic cap, enlarged necrotic core, and rich in macrophages resemble the main features of vulnerable plaques, whereas increased contents of collagen deposition and smooth muscle cell (SMC) proliferation favor plaque stability [17]. UII is widely expressed in the heart, blood vessels, and monocytes/macrophages thereby playing an important role in cardiovascular diseases [7,18]. UII can stimulate macrophages to lade more oxidized lipids and form foam cells and possibly act as a pathogenic factor of vulnerable plaques [10,19]. UII also plays a crucial role in regulating angiogenesis through mediating macrophage functions [20,21,22,23,24]. Another characteristic of vulnerable plaques is intra-plaque aberrant neovessel sprouting, but it is unknown if UII takes part in the angiogenesis. To examine the hypothesis of whether UII plays any role in atherosclerosis initiation, advanced plaque vulnerability, and lesional regression, we were determined to infuse UII into hypercholesterolemic rabbits which develop different stage atherosclerotic lesions.

Rabbits are cholesterol-sensitive animals and easily develop advanced atherosclerosis with a long-term high-cholesterol diet (HCD). Furthermore, advanced atherosclerotic lesions can be induced by long-term high-cholesterol diet feeding in wild-type rabbits or by angiotensin II infusion in Watanabe heritable hyperlipidemic rabbits [25,26]. The aims of this study include: (1) to investigate the effect of UII on atherogenesis in rabbits; (2) to clarify the effect of UII on aortic plaque vulnerability and coronary plaque formation in the mid to late stage atherosclerotic atheromatous plaque in rabbits; (3) to explore the possible mechanisms by which UII affects plaque vulnerability; and (4) to show the effect of UII on atherosclerotic plaque regression. All results suggested that UII may be a risk factor in all stages of atherosclerosis.

## 2. Results

### 2.1. UII Promotes Atherosclerotic Fatty Streak Formation

In male rabbits, six weeks of UII infusion slightly promoted the initiation of atherosclerosis; however, the difference is not significant (Appendix A). Therefore, in the female rabbit experiment, the UII infusion time was extended to 12 weeks. Compared to sham-operated rabbits, the E2 levels were significantly decreased in OVX rabbits (Figure 1D). Aorta intimal Sudan IV staining showed that twelve weeks of a 0.3% cholesterol diet successfully induced early atherosclerotic plaque formation in female rabbits. OVX significantly promoted the formation of atherosclerosis in rabbits, and the lesion area was about 3 times that of the vehicle group (Figure 1B). For microscopic immunohistochemical quantitative analysis, the aortic arch sample was sectioned and stained as previously described 1. Histological staining showed that UII infusion significantly increased the microscopic atherosclerotic lesion area by accelerating foam cell formation, promoting lipid accumulation and plaque progression (Figure 1C,E). Macrophage or smooth muscle cell-derived foam cells remain to be the two main cellular components in the early atherosclerotic lesion. UII chronic infusion selectively affected the macrophage-derived foam cell formation and showed less effectiveness in regulating smooth muscle cell function in ovariectomized rabbits (Figure 1G,H).

### 2.2. Infusion of UII Infusion Changes Aortic Plaque Cellular Components

In the advanced plaque experiment, both groups developed hyperlipidemia induced by HCD feeding. Aortic atherosclerosis was present in both groups but there was no significant difference between the two groups in terms of both gross lesions in the aorta (Figure 2B). Microscopic analysis of the lesions showed that compared with the control group, the macrophage-positive stained area was increased by 113% in the UII group (*p* = 0.0364) while the contents of collagen (32%) and SMCs (15%) were simultaneously decreased although not statistically significant (n.s.) (Figure 2C). Long-term HCD induced complicated plaque in rabbits, UII tends to enhance the progression of advanced lesions, including more intra-plaque cholesterol clefts, calcification, and even lipid pool (Appendix A). UII infusion did not change plasma lipid levels and body weight in male rabbits during the whole experiment (Appendix A).

### 2.3. Chronic Infusion of UII Increases Aortic Plaque Vulnerability and Promotes Intra-Plaque Angiogenesis

The finding that the lesions of the UII group were characterized by more macrophages and fewer SMCs and collagens prompted us to investigate the degree of plaque vulnerability (Figure 2C). The plaque vulnerability index of UII-infused rabbits was significantly higher than that in the control (Figure 2C). These results suggested that UII infusion may promote advanced plaque formation. Intra-plaque abnormal angiogenesis is the main hallmark of advanced vulnerable plaque and CD31 staining is well used to show neovessels [27,28]. We also investigated this issue using immunohistochemical (IHC) staining using CD31 Ab. Among ten UII-infused rabbits, eight of them showed blood vessel sprouting detected by CD31 staining whereas no intra-plaque neovessel was found in the control group (Figure 2D).

### 2.4. UII Infusion Accelerates Coronary Atherosclerosis

Seven blocks of the hearts were used for histological analysis of coronary plaques (Figure 3A). The coronary lesions were found in all blocks and expressed as coronary lumen stenosis (Figure 3C). Though the plaque size and lumen stenosis varied among rabbits, the UII group apparently showed a tendency to have more lesions (except block II). UII treatment significantly increased coronary stenosis in blocks III (127% increase) and VI (95% increase). Similar to aortic lesions, UII infusion decreased collagen contents and increased the proportion of macrophages in the coronary lesions (Figure 3D).

### 2.5. UII Infusion Exacerbates Carotid and Subclavian Arteries

Compared to control rabbits, UII infusion significantly exacerbated the progression of aortic branch lesions (carotid and subclavian arteries) (Figure 4A). Compared with the controls, the atherosclerotic lesion size of the branch arteries was obviously increased by 69% in UII-infused rabbits (Figure 4A). UII also increased the microscopic lesions compared to the control group even though the difference did not reach statistical significance (Figure 4B).

### 2.6. UII Delays the Regression of Atherosclerotic Lesion

In the atherosclerotic regression study, atherosclerosis was successfully induced by 24 weeks of HCD feeding, normal chow diet was feeding for another 8 weeks for plasma cholesterol recovery to normal level (Appendix A). Twelve weeks of UII infusion slightly delayed the regression of atherosclerosis by maintaining a higher lesion area (Figure 5B). The IHC analysis showed that UII infusion changed the plaque cellular components by increasing macrophages derived foam cells ratio but decreasing collagen and SMCs proportion (Figure 5C). These results suggested that UII may be also involved in the regression of atherosclerotic lesions even in plasma lipids in well-controlled animals.

### 2.7. UII Stimulates ROS Generation and Activates HIF-1α/VEGF-A Pathway

The cultured cell experiments showed that UII did upregulate the expression of the NADPH oxidase type 2 (NOX2) protein in macrophages (Figure 6A). The UII-induced upregulation could be partly blocked by urantide, an inhibitor of UII (Figure 6A). As NOX2 is a key regulator of ROS generation, we further examined the ROS production using immunofluorescence staining and found that UII treatment significantly increased ROS generation in macrophages (Figure 6B). In addition, mRNA and protein levels of hypoxia-inducible factor (HIF)-1α and vascular endothelial growth factor (VEGF)-A were also upregulated by UII treatment in macrophages (Figure 6D). In the same vein, UII-induced high expression levels of HIF-1α and VEGF-A in macrophages could be inhibited in the presence of urantide. Moreover, as shown in Figure 6E, the immunofluorescent staining also confirmed the high expression of HIF-1α on macrophages after UII treatment. To examine whether UII is involved in angiogenesis which was found in the lesions above, we performed a Matrigel-based angiogenesis assay. Our in-house experiment optimized the time for human umbilical vein endothelial cells (HUVECs, CRL-1730) seeding for tubule formation analysis and 4 h were selected for all experiments (Appendix A). Pretreated HUVECs with the vehicle, UII or UII plus urantide were seeded into Matrigel for analysis of tubule formation. UII treatment increased the total vessel length in Matrigel (Figure 6G,H).

## 3. Discussion

In this study, it was found that UII infusion may promote the initiation of atherosclerosis in both male and ovariectomized female rabbits. Our previous study also found that UII has no obvious effect on the lesions in female rabbits [10]. We speculate that the effect of estrogen may mask the effect of UII on atherosclerosis in females. The female sex shows protective effects in the incidence and complications of atherosclerosis, and this may partly be through 17β-estradiol (E2), a potent endogenous estrogen [29,30]. Postmenopausal women are less different from men in the development of atherosclerosis. In this study, we first reported that UII promotes the progression of atherosclerosis in ovariectomized rabbits. Estrogen deficiency in postmenopausal women may reduce the anti-inflammatory and antioxidant capacity of arteries, thereby contributing to atherosclerosis formation. UII and its receptor expression can be upregulated by inflammatory stimuli. In postmenopausal women, whether the inflammation-prone state induces vasoactive peptide production, such as UII, and plays a role in their cardiovascular disease, needs further clinical data to confirm. The results of this study suggested that vasoactive peptides, such as UII, may be involved in the modulation of atherosclerosis progression in postmenopausal females like that in males.

In the current study, the effects of long-term chronic infusion of UII on advanced atherosclerosis were investigated in cholesterol-fed male rabbits. Infusion of UII led to several changes in the lesions of the aortic, carotid/subclavian arteries, and coronary arteries compared with the control group. First, UII infusion did not alter the plasma lipids and general state of rabbits since all plasma lipid parameters and body weights were similar in the two groups. UII infusion did not change the aortic surface gross lesions compared with the control group, which is different from the short-term study reported previously [10,13]. However, this finding may not be surprising because the whole surface of the aorta was almost totally covered by the lesions in both groups and a longer duration of hypercholesterolemia may overwhelm the effect of UII on the gross lesion expansions, which is different from early lesions induced by short-term hypercholesterolemia reported previously [10,13]. In support of this assumption, aortic branches including carotid and subclavian arteries developed more lesions in the UII group than in controls. It should be pointed out that these branches in hypercholesterolemic rabbits rarely developed the lesions but carotid atherosclerosis is much more relevant to human situations [31,32].

The second important finding derived from the current study is that the lesions of the UII group were histologically characterized by enhancement of vulnerability, including more macrophages and less smooth muscle cells and collagen contents along with the increased vulnerable index. One of the possible mechanisms for these histological changes may be owing to the enhanced emigration of monocytes from the circulation because UII induces VCAM-1 and ICAM-1 expression on the endothelium [10,33]. These accumulated macrophages may worsen the lesion progression by the production of various bioactive substances such as ROS. This notion was supported by our in vitro studies showing that UII treatment induces ROS generation.

Finally, we found that the lesions of the UII group were rich in angiogenesis compared with the control group. The aortic intra-plaque pathological angiogenesis in the UII group suggests that it may promote advanced atherosclerosis by regulating abnormal angiogenesis. Although the pathophysiological significance of these neovessels in the lesions remains unclear, many studies reported that intra-plaque blood vessels destabilize plaque stability because recurrent intra-plaque hemorrhage can lead to plaque rupture [34,35,36]. UII may mediate the angiogenesis through several possible pathways. The plaque microenvironment is subjected to hypoxia which may facilitate the angiogenesis through the upregulation of VEGF and HIF-1α expressed by macrophages as we found in the current study. In addition to the direct role of UII, the over-production of ROS may be also involved in angiogenesis [37,38]. As a regulator for ROS production, upregulated NOX2 induced by UII may also impair macrophage functions and/or promote angiogenesis [38,39]. In such a circumstance, UII may generate a vicious circle in the lesions: the more macrophages accumulate, the more ROS is produced.

We also found that UII infusion led to severe coronary atherosclerosis compared with the control group. Unlike the aortic lesions, coronary atherosclerosis is difficult to induce in most experimental animal models. The coronary arteries of rodents are too small to resemble humans, and the high cost of big animals such as pigs and monkeys makes it difficult to be widely used. The middle size of rabbits shows some advantages in coronary atherosclerosis research [25,40]. It will be interesting to investigate whether these coronary lesions can cause myocardial infarction in the future.

In this study, UII infusion also delayed the plaque regression by changing the lesion cellular components. It was reported that UII can suppress ATP-binding cassette transporter A1 expression and reduce cholesterol efflux of macrophages [41,42]. In another way, the role of UII in enhancing the production of ROS and the progression of inflammation may also be involved in inhibiting atherosclerotic regression [43,44,45].

## 4. Materials and Methods

### 4.1. Animals

In animal experiment I, to investigate the role of UII on a fatty streak (early stage of atherosclerosis), male rabbits were subcutaneously infused with either UII (dissolved in saline, 5.4 μg/kg/h) or saline (n = 6 for each group) via osmotic pumps for 6 weeks while they were fed an HCD (Appendix A). Endogenous estrogen is a well-defined anti-atherogenic factor [29]. Therefore, female rabbits underwent ovariectomy surgery to simulate the postmenopausal state [46,47]. After a sham operation or ovariectomy and following two weeks recovery, rabbits were divided into three groups: sham with saline infusion, ovariectomy with saline, and ovariectomy with UII infusion (5.4 μg/kg/h), and continue 12 weeks HCD feeding to induce atherosclerosis (Figure 1A). Experiment II was designed to study the role of UII on advanced atherosclerosis pathogenesis and coronary plaque formation. After two weeks of HCD (supplemented with 3% soybean oil and 0.3% cholesterol) feeding, those rabbits with plasma cholesterol less than 300 mg/dL were excluded. The remained rabbits were subcutaneously infused with either UII (dissolved in saline, 5.4 μg/kg/h) or saline (n = 10 for each group) via osmotic pumps for 24 weeks while they were fed an HCD to induce advanced plaque (Figure 2A). For the plaque regression study in experiment III, male rabbits were fed HCD for 24 weeks to induce atherosclerosis, and then change to chow diet feeding until the plasma cholesterol dropped to the normal level (about 8 weeks), either vehicle or UII was infused another continued 12 weeks to analysis the plaque regression (Figure 5A). Synthetic human UII (Cat #950263) and urantide (Cat #950264) were purchased from GL Biochem (Shanghai, China) Corporation. Ltd. The osmotic mini-pumps were provided by DURECT Corporation (ALZET Model 2006, Cupertino, CA, USA). The dose of UII for infusion was selected based on our previous studies [10,13]. Osmotic mini-pumps were changed every six weeks. The male wild rabbits were provided by the Laboratory Animal Center of Xi’an Jiaotong University. The experimental protocols of animal study were reviewed and approved by the Laboratory Animal Administration Committee of Xi’an Jiaotong University (No. 2020-1045).

### 4.2. Plasma Lipids, Estradiol, and Blood Pressure Measurement

A blood sample was drawn from the medial auricular artery of a fasting (16 h) rabbit every four weeks. Plasma total cholesterol, triglycerides, high-density lipoprotein-cholesterol, and low-density lipoprotein-cholesterol were measured by commercial kits (BioSino, Beijing, China). Estradiol was measured by electrochemiluminescence assay according to the instructions (Roche Diagnostics GmbH, Mannheim, Germany). The body weights of the rabbits were also weighed every 4 weeks. The blood pressure (BP) and heart rate were measured using PowerLab (ADInstruments, New South Wales, Australia) via medial auricular artery cannulating before rabbit sacrifice as previously described [10,25].

### 4.3. Quantitative Analysis of Rabbit Aortic Atherosclerotic Lesions

After 24 weeks of UII or saline infusion, rabbits were sacrificed by sodium pentobarbital overdoses. The entire rabbit aorta tree along with the heart was collected and trimmed by removing all adipose and connective tissues. The entire aortic tree was divided into four parts for further analysis (Figure 2A): (I) the heart was removed for coronary analysis (part 1); (II) a ring of 2 to 3 mm was cut at the end of the aortic arch and embedded in OCT for plaque vulnerability analysis (part 2); and (III) the remaining aorta (part 3) along with branches including carotid arteries and subclavian arteries (part 4) was fixed on the cork board with stainless pins and cut open longitudinally using scissors from the aortic arch to the iliac artery to expose the aortic surface and then fixed in buffered formalin solution. The aortic tree was stained with Sudan IV for the quantitative analysis of sudanophilic gross atherosclerotic lesion area. Sudan IV-stained aorta was photographed and the lesion area was calculated by an image software (Mitani Co., Ltd., WinRoof 6.5, Tokyo, Japan). For microscopic lesional quantitative analysis, the arch was cut into 10 cross segments and then embedded in paraffin for sectioning. Histopathological staining, including hematoxylin and eosin (H&E), Elastic van Gieson (EVG), and Masson trichrome, was conducted in cross sections (4 μm) of the aortic arch. For atherosclerotic cellular components analysis, IHC staining of macrophages and SMCs was performed as previously described [48]. Antibodies (Abs) against macrophages (Cat#M0633, 1:500; Dako, Glostrup, Denmark), alpha-actin (SMCs, Cat#MS-742, 1:200; Thermo, Fremont, CA, USA) and endothelial cells (CD31, Cat#GC-0322; Jinqiaoyatu, Beijing, China) were used for these IHC staining. The goat anti-mouse conjugated with HRP secondary Ab was from Abcam (1:1000, Abcam, Cat#ab6789, Cambridge, UK). The microscopic lesions on the EVG-stained slides and IHC-staining positive areas were quantified using the image analysis software described above.

### 4.4. Analysis of Aortic Intra-Plaque Angiogenesis

Intra-plaque neovessels were marked by CD31 (an endothelial cells marker) positive staining in order to evaluate the neovascularization of the advanced atherosclerosis lesion. All slides stained with CD31 Ab were observed and those with intra-plaque angiogenesis were counted and recorded.

### 4.5. Evaluation of Plaque Vulnerability

The vulnerability index was used to evaluate the plaque features of the aortic lesions. For this purpose, serial frozen sections (6 μm) were stained with oil red O for quantitation of plaque lipids and Masson trichrome for quantitation of collagen deposition [49]. IHC staining for macrophages and SMCs was also performed. The VI was calculated with the following formula: VI = (macrophage-positive area + lipid area)/(SMC-positive stained area + collagen-positive stained area) [13,17,50].

### 4.6. Quantification of Coronary Stenosis and Plaque Cellular Components

Coronary plaque characteristics analysis was conducted by following previously described methods [25,48,51]. To quantitate coronary lumen stenosis and plaque cellular components, the rabbit heart was cut into 7 blocks (blocks I–VII). All blocks were embedded in paraffin for the following sectioning (4 μm) and stained with H&E and EVG to quantify coronary lumen stenosis (%) with the following formula: S = [I ÷ ((D/2)^2^ · π)] × 100%, I = area of intimal plaque lesion; D = transformed lumen diameter, D = Perimeter/π; and S = degree of coronary artery cross-sectional stenosis (Figure 3A). The lumen stenosis (%) was quantified using the method described above. Coronary plaque macrophages and collagen components were also analyzed.

### 4.7. Reactive Oxygen Species (ROS) Assay

The macrophage cell line (RAW264.7) was used for the analysis of ROS production. Cells were cultured in DMEM medium (Cat#12800-017; Hyclone, South Logan, UT, USA) supplemented with 10% fetal bovine serum (Cat#2177370; Gibco, Grand Island, NY, USA) in an incubator at 37 °C, 5% CO_2_ in the humid atmosphere. Cultured cells were divided into three groups: (1) vehicle: same volume solvent was added; (2) UII group: 100 nM UII for 24 h; (3) UII + Urantide group: cells were co-treated with 100 nM UII plus 100 nM urantide for 24 h. After treatment with UII or urantide, the cells in the 12-well culture dishes were loaded with 10 μM of 2′,7′-dichlorofluorescein diacetate (Cat#S0033; Beyotime, Shanghai, China) and cultured for another 0.5 h. Cultured cells were observed under a fluorescent inverted microscope in a dark room. ROS levels were quantified as the mean fluorescence intensity by image software.

### 4.8. Quantitative Real-Time Reverse Transcription PCR (qRT-PCR)

RAW264.7 cells, a macrophage line, were cultured to reach 70 to 80% confluence in 12-well culture dishes and incubated for 24 h with UII or urantide. The cells were then treated with Trizol reagent (Cat#CW0580; CWBIO, Beijing, China) to extract total RNA and then synthesize cDNA with a reverse transcription kit (Cat#AG11706; Accurate Biology, Changsha, China). qRT-PCR was conducted by using an SYBR kit (Cat# AG11701; Accurate Biology) to amplify the prepared cDNA samples. Primer sequences: HIF-1α (Forward: GAATGAAGTGCACCCTAA-CAAG; reverse: GAGGAATGGGTTCACAAATCAG); VEGF-A (forward: CTGTGCAGGCT-GCTGTAACG; reverse: GCTCATTCTCTCTATGTGCTGGC); GAPDH (Forward: CATCCGTAAAGACCTCTATGCCAAC; reverse: ATGGAGCCA-CCGATCCACA). The target gene expression levels were calculated using the comparative Ct method for statistics, X = 2^−ΔΔCt^.

### 4.9. Western Blotting

In the same experiments as above, total proteins were extracted from cultured RAW264.7 cells, and the concentrations were determined by a BCA Protein Assay Kit (Cat#PA115-01; TianGen Biotech, Beijing, China). For Western blotting (WB) analysis in cell experiments, Abs against HIF-1α (Cat#BF0593; 1:1000; Affinity, Cincinnati, OH, USA), VEGF-A (Cat #AF5131; 1:1000; Affinity, Cincinnati, OH, USA), NOX2 (Cat#DF6520; 1:1000; Affinity, Cincinnati, OH, USA) and GAPDH (Cat#5174; 1:2500; CST, Danvers, MA, USA), followed goat-anti-rabbit (Cat#7074; 1:2500; CST, Danvers, MA, USA) or mouse HRP-linked secondary antibody (Cat#ab6789,1:5000; Abcam, Cambridge, UK) were used. Protein samples (20 μg) were fractionated on 8% SDS-PAGE and transferred to a membrane and probed with Abs against NOX2, HIF-1α, VEGF-A, and GAPDH. The intensity of the protein bands was quantified as mean OD values by Image J software (Boao Yijie, Beijing, China).

### 4.10. Tubule Formation Assay

For the tubule formation assay, Matrigel (Cat#0827015; Noven Pharmaceuticals, Inc, Shanghai, China) was placed on crushed ice and slowly thawed at 4 °C overnight. Meanwhile, the pipettes and 96-well plates were precooled at −20 °C. The next day, the melted Matrigel was added to precooled plates and placed in the cell culture incubator for approximately 35 min to promote gelation. HUVECs (CRL-1730) were prepared by pretreatment with UII (100 nM), UII (100 nM) plus urantide (100 nM), or their solvent for 24 h before seeding. The pretreated HUVECs were then digested with trypsin and counted for the following analysis. HUVECs were seeded into Matrigel (50 μL per well) precoated plates at 2 × 10^4^ cells/well. Cultured HUVECs were observed under an inverted microscope and five high-power fields were randomly photographed and the tubular formation was analyzed by using Angio Tool 640.6a software (SAIC, Birmingham, MI, USA).

### 4.11. Statistical Analysis

All quantitative data are shown as mean ± SEM. For normally or approximately normally distributed data, a Student’s *t*-test was performed. One-way ANOVA analysis was performed for a comparison of three or more groups. For non-normal data analysis, the nonparametric test was performed. For analysis of intra-plaque angiogenesis positive sample number, Fisher’s exact test was used. Statistical analysis was conducted by Prism 9.0 software (GraphPad Software, CA, USA). The *p* < 0.05 was considered a significant statistical difference.

## 5. Conclusions

Our results showed that UII plays multiple roles in the pathogenesis, progression, and regression of atherosclerosis. Importantly, UII upregulates the genes that are involved in ROS production and angiogenesis which destabilize the plaque vulnerability. Although molecular mechanisms are still not completely disclosed, these results indicate that targeting UII may become a new strategy for the prevention of plaque rupture in the future.

## Figures and Tables

**Figure 1 ijms-24-03819-f001:**
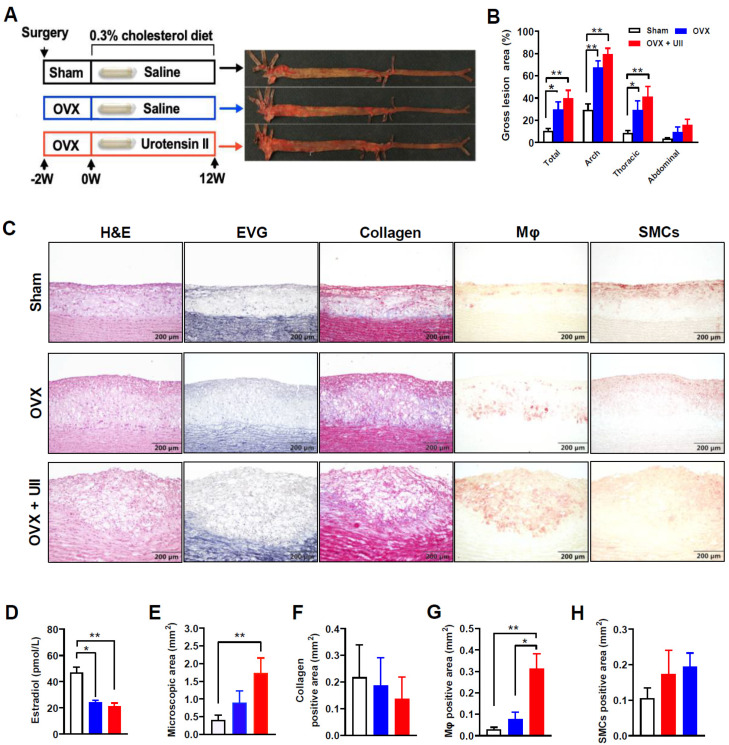
UII infusion aggravates atherosclerosis in ovariectomized female rabbits. (**A**) Study design: UII (5.4 μg/kg/h) or saline was infused via osmotic pumps into ovariectomized rabbits with a 0.3% cholesterol diet for 12 weeks. The en face lesions were visible after Sudan IV staining. (**B**) Quantifications of the lesional area of the whole aortic tree. (**C**) Representative images of immunohistochemical stainings. (**D**) Estrogen levels of sham and ovariectomized female rabbits. (**E**–**H**) Quantifications of immunohistochemical staining. All data are expressed as mean ± SEM. All data were tested for normality and equal variance. If passed, one-way analysis of variance (ANOVA) followed by Tukey post hoc test for comparisons among >2 groups. Otherwise, nonparametric tests (Kruskal–Wallis test followed by Dunn’s post hoc test) were used. *n* = 6–10 per group. * *p* < 0.05 and ** *p* < 0.05 vs. vehicle group.

**Figure 2 ijms-24-03819-f002:**
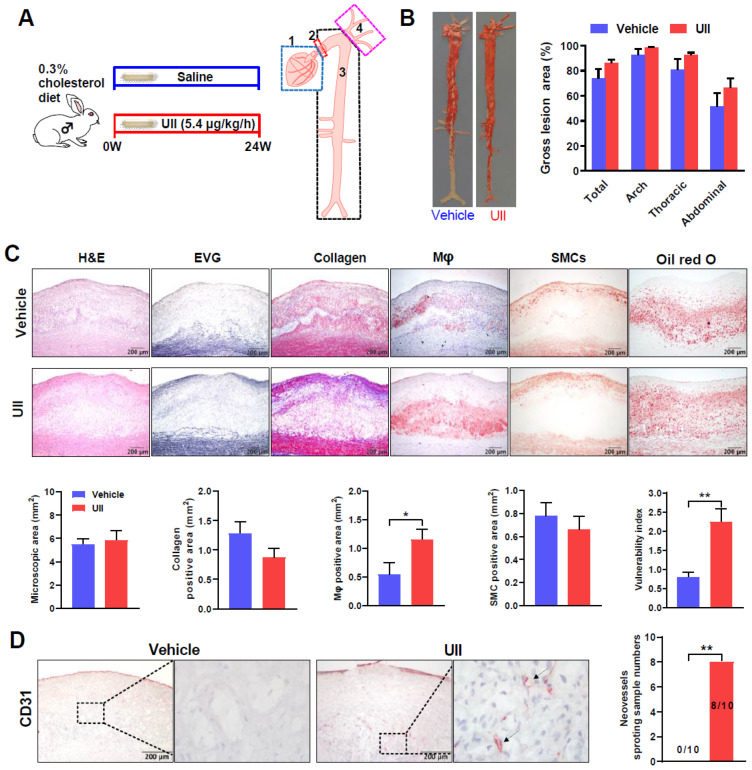
Schematic illustration of the advanced plaque experiment design and gross lesion analysis. (**A**) Rabbits were fed an HCD and were infused with either UII (5.4 μg/kg/h) or saline via osmotic mini-pumps for 24 weeks (left panel). Aortic trees along with the heart were collected and divided into 4 parts for the following analysis: coronary plaque analysis (1); an aortic ring for calculating plaque vulnerability index (2); remaining aorta tree for analyzing aortic lesions (3), and carotid and subclavian arteries of the aortic branches (4) (right). (**B**) The representative images of Sudan IV-stained aortas from each group (left panel). Quantification of the sudanophilic area of the whole aortas was shown (right panel). (**C**) Microscopic analysis of the aortic lesions. Representative micrographs stained by H&E, EVG, collagen, macrophages (Mφ), SMCs, and Oil red O from each group (upper panel). Quantifications of microscopic atherosclerotic area, collagens, Mφ, and SMCs and vulnerability index of aortic arch plaques (lower panel). * *p* < 0.05, ** *p* < 0.01 vs. vehicle group. (**D**) Analysis of intra-plaque angiogenesis. Neovessels in the aortic plaques were detected by CD31 staining (arrows). The number of animals with intra-plaque neovessels was calculated. Fisher’s exact test was performed to compare the intra-plaque angiogenesis between the two groups. All data are presented as mean ± SEM, *n* = 10/group.

**Figure 3 ijms-24-03819-f003:**
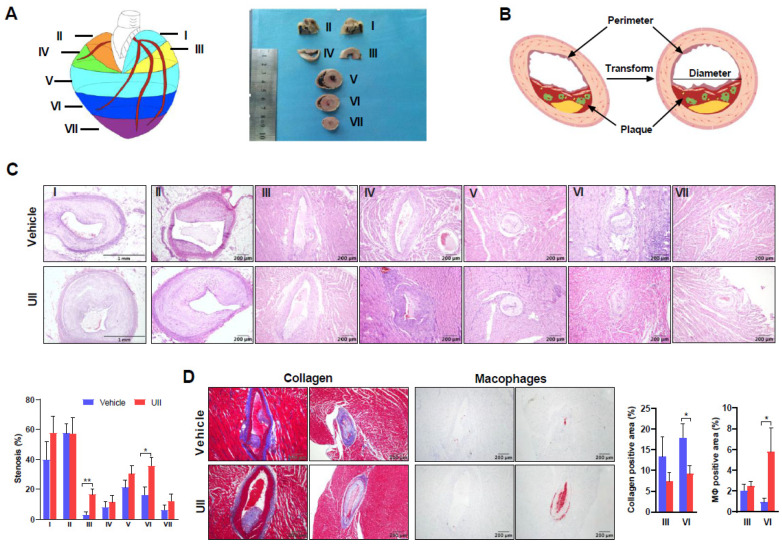
Analysis of coronary atherosclerosis. (**A**) Schematic illustration of heart sectioning procedure. (**B**) Calculation method of coronary atherosclerotic lesions. (**C**) Representative figures of coronary atherosclerosis of seven blocks. The sections were stained with H&E stain of each group. Quantification of coronary arterial stenosis rate (%) (left panel). * *p* < 0.05, ** *p* < 0.01 vs. vehicle group. (**D**) The lesions (blocks III and VI) were either stained with Masson trichrome for collagen or immunohistochemically stained with anti-Mφ Ab (left panels). Positively stained areas were quantified (right panel). n = 3–10/group, all data are presented as mean ± SEM.

**Figure 4 ijms-24-03819-f004:**
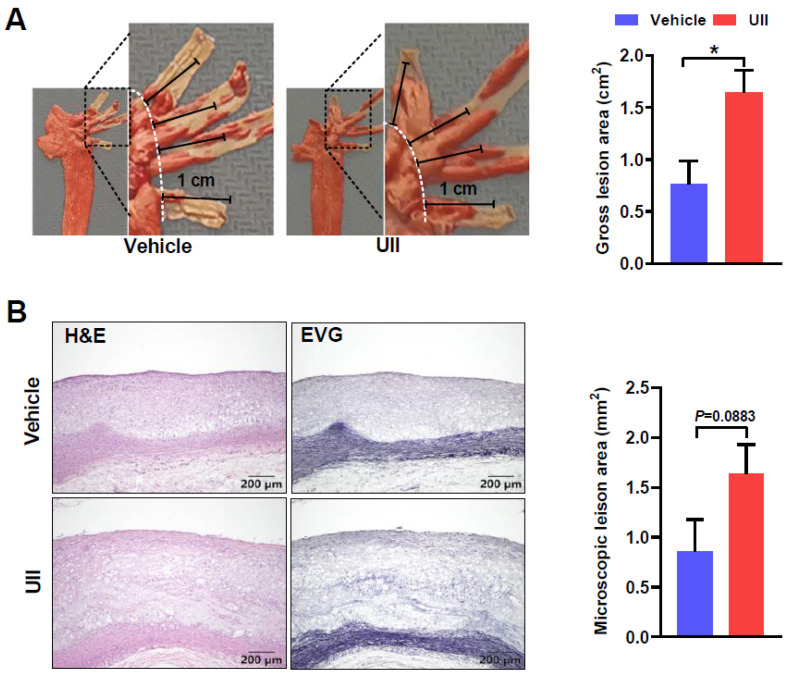
Analysis of the lesions of the common carotid and subclavian arteries. (**A**) Analysis of gross lesion areas. Representative images of four branches (containing right subclavian and common carotid arteries, left subclavian and common carotid arteries) stained with Sudan IV (left panel). The plaque area distributed within 1 cm extending upward from the root was measured and the total areas of four branches were shown (right panel). (**B**) Analysis of the microscopic lesions. These branches were stained with either H&E or EVG stain (left panel). The lesion size was measured using EVG-stained sections (right panel). *n* = 10/group, all data are presented as mean ± SEM. * *p* < 0.05.

**Figure 5 ijms-24-03819-f005:**
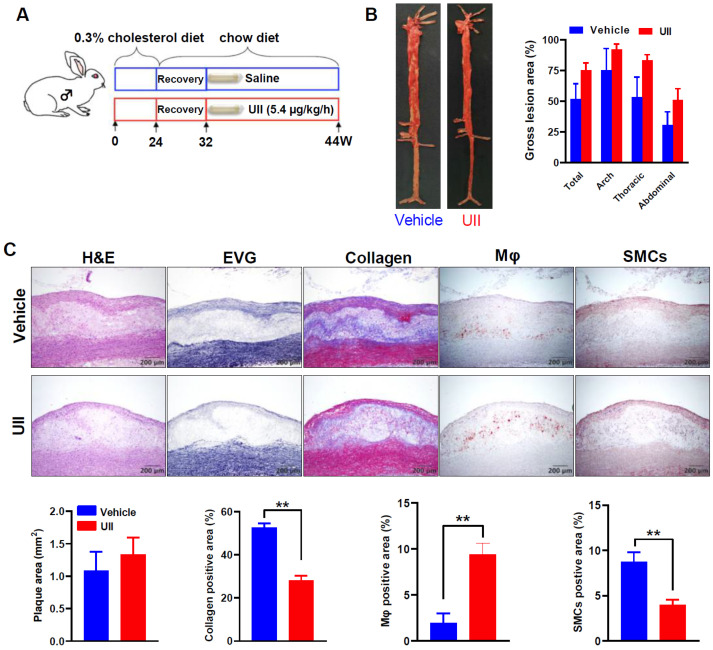
UII delays the regression of atherosclerosis in rabbits. (**A**) Study design: advanced atherosclerosis was induced by HCD feeding for 24 weeks, then changed to chow diet for 8 weeks recovery for plasma cholesterol decreased to a normal level, and UII (5.4 μg/kg/h) or saline was infused for another 12 weeks. (**B**) Representative images of aortas after Sudan IV staining and their gross lesional quantification. (**C**) Representative microscopic images and quantifications of immunohistochemical staining. All data are expressed as mean ± SEM. *n* = 5–6 per group. ** *p* < 0.05 vs. vehicle group.

**Figure 6 ijms-24-03819-f006:**
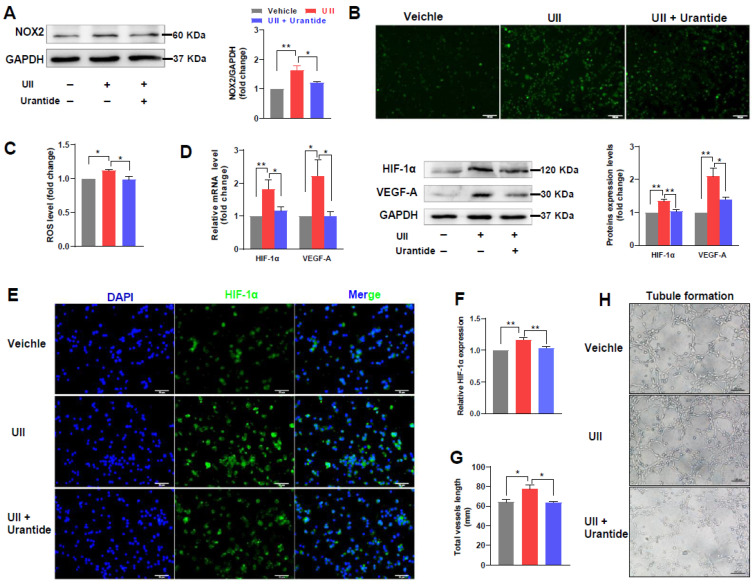
Determination of UII effects on the cultured cells. (**A**) NOX2 expression after being treated with 100 nM UII with or without urantide for 24 h in cultured macrophages. (**B**,**C**) ROS synthesis in macrophages was determined by the detection of the DCFH-DA fluorescent intensity. Representative images were shown in (**B**) and fluorescent intensity was quantified and shown in (**C**). Scale bar, 100 μm. (**D**) qRT-PCR (left) and WB (right) analyses of HIF-1α/VEGF-A expression in macrophages after UII with or without urantide treatment for 24 h. UII treatment significantly increased the expression of these two genes but was inhibited by the presence of urantide. Western blotting confirmed that UII upregulated HIF-1α/VEGF-A expression and this upregulation could be partly blocked by urantide. (**E**) and (**F**) HIF-1α expression in macrophages was detected by fluorescence microscopy and green fluorescence was used to label HIF-1α, scale bar, 50 μm (**E**). UII upregulated HIF-1α expression and could be partially blocked by urantide. Quantification of the relative intensity of HIF-1α fluorescence (**F**). (**G**) and (**H**) UII stimulated angiogenesis in Matrigel with seeded HUVECs after 4 h of culture. Quantification of tubule formation, scale bar, 100 μm. Data are shown as mean ± SEM (three to five biological repeats). * *p* < 0.05, ** *p* < 0.01 vs. vehicle group.

## Data Availability

All data were included within the article.

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
