# Peer review of "Urotensin II Enhances Advanced Aortic Atherosclerosis Formation and Delays Plaque Regression in Hyperlipidemic Rabbits"

_ijms, 2023, doi:10.3390/ijms24043819_

Round 1
Reviewer 1 Report
Yu et.al investigated the role of urotensin II on the inititaion, progression of atherosclerosis, as well as regression of established atherosclerosis using rabbits fed high cholesterol diet and cells (macrophages and endothelial cells) related to atherosclerosis. They observed that choronic adminstation of urotensin II deteriorates all stages of atherosclerosis in vivo. In vitro studies further showed that urotensin II increases ROS production in macrophages and also promotes angiogenesis of endothelial cells. As such, they stated that urotensin II enhances advanced atherosclerosis formation and delays establisehd atherosclerosis regression in vivo.
In general, this study was well designed and also showed some novel findings on the role of urotensin II on atherosclerosis develpment, especially the regression of establisehd atherosclerosis. My minor concers are as follows.
1. Introduce a bit more information on urotensin II biology, especially in the context of atherosclerosis. For example, in paitents with atherosclerotic diseases, how many urotensin II occures in the blood circulation.
2. Please show the reason on the dosage of urotensin II in vivo and also in vitro studies.
3. Does urotensin II undergoes metabolism in vivo? If so, whether its metabolites are responsibe for the effects observed in vivo and in vitro.
4. Does the sample size (3-10 per group) is enough to evaluate the effect of urotensin II on atherosclerosis formation, progression and also regression.
5. There is a lack linking between in vivo and in vitro observations. Can the authors show some data or discussions on this point, which may further support their conclusions.
6. There is no standardized index to quantify the extent of plaque vulnerability. How about the features for plaque vulnerability through the manuscript?
7. Last but not least, line 30: delay the progression of atherosclerosis was assumed to be delay the regresion.
Reviewer 2 Report
The current original article evaluate the role of urotensin II on atherosclerosis in an experimental article. It is well written and can be of further clinical implications. Comments:
- At the end of introduction, it should be clear specified the aim of the current study.
- In fig 1, there is no I figure.
- The material and methods should be placed before results.
- The last paragraph of discussion should be separated to conclusion and no references should be added there. Maybe shortened.
- Maybe introduced a paragraph for making the translation to clinical practice.
- Moderate English changes.
Reviewer 3 Report
UrotensinII is related to cardiovascular diseases such as AS. In this manuscript the authors use animal model to determine the role of Urotensin II in AS progression. The results show that UII has pro-angiogenic effect but delay the progression of the AS via increasing NOX2 and HIF1a/VEGF-A pathway. Below are my comments on the manuscript.
1. Please pay attention to English grammar, spelling, and sentence structure and make all changes clearly.
2. There are many studies on medications for NAFLD. Which stage of NAFLD do you think kaempferol is mainly involved in treatment and what are its advantage? please give an explanation.
Round 2
Reviewer 2 Report
Thank you for considering my suggestions. I think the manuscript is suitable for publication.